



# Statistical analysis of the long-range transport of the 2015 Calbuco volcanic eruption from ground-based and space-borne observations

**Nelson Bègue[1], Lerato Shikwambana[2,3], Hassan Bencherif[1,3], Juan Pallotta[4], Venkataraman Sivakumar[3], Elian Wolfram[4], Nkanyiso Mbatha[5], Facundo Orte[4], David Jean Du Preez[6], Marion Ranaivombola[1], Stuart Piketh[7] and Paola Formenti[8]**

[1] Laboratoire de l'Atmosphère et des Cyclones, UMR 8105 CNRS, Université de la Réunion, Reunion Island, France.

[2] Space science Division, South African National Space Agency, Hermanus 7200, South Africa

[3] School of Chemistry and Physics, University of KwaZulu-Natal, Private Bag X54001, Durban 4000, South Africa

[4] Centro de Investigaciones en Láseres y Aplicaciones, UNIDEF (CITEDEF-CONICET), 5 Villa Martelli, B1603ALO, Buenos Aires, Argentina

[5] University of Zululand, Department of Geography, KwaDlangezwa, 3886, South Africa

[6] Department Geography, Geoinformatics and Meteorology, University of Pretoria, Pretoria, 0002, South Africa

[7] North-West University, Potchefstroom, South Africa

[8] Laboratoire Interuniversitaire des Systèmes Atmosphériques, UMR CNRS 7583, Université Paris-Est-Créteil, Université de Paris, Institut Pierre Simon Laplace, Créteil, France

Correspondence to: N.Bègue (nelson.begue@univ-reunion.fr)

## Abstract

This study investigates the influence of the 2015 Calbuco eruption (41.2°S, 72.4°W; Chile) on the total columnar aerosol optical properties over the Southern Hemisphere. The well-known technic of sunphotometry was applied for investigation of the transport and the spatio-temporal evolution of the optical properties of the volcanic plume. The CIMEL sunphotometer measurements performed at 6 South American and 3 African sites were statistically analyzed. This study involves the use of the satellite observations and a back-trajectory model. The passage of the Calbuco plume is statistically detectable on the aerosol optical depth (AOD)



observations obtained from sunphotometers and MODIS. This statistical detection confirms that the majority of the plume was transported over the northeastern parts of South America and reached the South African region one week following the eruption. The plume has impacted to a lesser extent the southern parts of South America. The highest AOD anomalies were observed over the northeastern parts of the South America. Over the South African sites, the AOD anomalies induced by the spread of the plume were quite homogeneously distributed between the east and west coast. The optical characteristics of the plume near source region was consistent with a bearing-ash plume. Conversely, the remote sites to the Calbuco volcano were influenced by ash-free plume. The optical properties discuss on this paper will be used as inputs for numerical models for further investigation on the ageing of the Calbuco plume in a forthcoming study.

## 1. Introduction

Given that the major volcanic eruptions have the potential to inject large amounts of sulfur into the stratosphere, they are considered as one of the main sources of stratospheric sulfur (Carn et al., 2015; Thomason et al., 2007). Sulphate aerosols are formed in the volcanic plume by aqueous and gaseous oxidation of sulfur dioxide ($SO_2$) and the subsequent nucleation and accumulation of particles and droplets (Watson and Oppenheimer, 2000). Volcanic emissions may have a significant impact on the atmospheric composition and radiative budget (McCormick et al., 1995; Solomon et al., 1999, 2011). McCormick et al. (1995) showed that following a major volcanic eruption the increase aerosols loading can lead to significant warming of the middle atmosphere. For instance, a warming ranging from 1°C to 4°C was observed in the tropical stratosphere following the Pinatubo eruption in 1991 (Labitzke and McCormick et al., 1992; Young et al., 1994). Through the use of ground-based and satellite observations, various studies have shown that a significant ozone loss occurs following a major volcanic eruption (Hofmann and Oltmans, 1993; Solomon et al., 2005). The sulphate aerosols formed following these events provide surfaces for heterogeneous chemical reactions, which lead to ozone depletion (Tie and Brasseur, 1995; Solomon et al., 1996; Bekki et al., 1997).

Previous studies have also pointed out that moderate volcanic eruptions (i.e, volcanic explosive index between 3 and 5) can significantly modulate the stratospheric aerosol loading compared to the "background period" (i.e, free from the effects of a major volcanic eruption) (Haywood et al., 2010; Neely III et al., 2013). Both ground-based (Hofmann et al., 2009; Trickl et al., 2013; Zuev et al., 2017) and satellite (Vanhellemont et al., 2010; Vernier et al., 2011) observations suggest that the aerosol optical depth (AOD) of the stratospheric aerosol layer



between 20 and 30 km has increased by 4 - 10% per year since 2000. Through the use of the
Whole Atmosphere Community Climate Model (WACCM version 3), Neely III et al. (2013)
showed that the increase in AOD of the stratospheric aerosol layer is likely due to moderate
volcanic eruptions. Satellite observations confirm that the decadal increase in stratospheric
aerosol loadings are linked to a series of moderate volcanic eruptions that each injected around
a megaton of $SO_2$ in the lower stratosphere (Vernier et al., 2011). In spite of the fact that these
recurrent volcanic eruptions inject less $SO_2$ than major volcanic eruptions, they can impact the
atmospheric radiation budget. Furthermore, taking into account the stratospheric aerosol burden
in climate models has been shown to be necessary since their trend has led to a significant
counterbalance of the global warming, so called the global warming hiatus (Solomon et al.,
2011; Fyfe et al., 2013; Haywood et al., 2013; Ridley et al., 2014; Santer et al., 2014). The use
of the Canadian Earth System Model (CanESM2), Fyfe et al. (2013) revealed that the moderate
volcanic activity since 2000 has contributed to a reduction of global warming with an impact
of -0.07 ± 0.07 K. Based on the use of the coupled atmosphere ocean Earth System model
(HadGEM2-ES), Haywood et al. (2013) showed a global mean cooling around -0.02 to -0.03
K over the period 2008-2012 period. They showed that the eruptions may result the perceived
hiatus in global temperatures caused by the small cooling effect but do not appear to be the
primary cause. These previous studies highlight the importance of the stratospheric aerosol
burden in climate models and to pursue the analysis of the moderate volcanic activity.
The moderate volcanic eruptions are considered as the most influential events on the
stratospheric aerosol burden during the last decade. These moderate volcanic eruptions have
mainly been observed in the Northern Hemisphere (Bourassa et al., 2010; Clarisse et al., 2012;
Jégou et al., 2013; Kravitz et al. 2010; Sawamura et al., 2012). The Southern Hemisphere was
mainly affected by three moderate eruptions since 2010: (1) the Puyehue-Cordon Caulle
(40.3°S, 72.1°W; Chile) in June 2011 which emitted 0.2 Tg of $SO_2$ into the upper troposphere
and lower stratosphere (UTLS); Clarisse et al., 2013; Theys et al., 2013; Koffman et al., 2017),
(2) the Kelut eruption (7.5°S, 112.2°E; Indonesia) in Feburary 2014, which injected 0.1-0.2 Tg
of $SO_2$ into the stratosphere (Kristiansen et al., 2015; Vernier et al., 2016), and (3) the Calbuco
eruption (41.2°S, 72.4°W) in April 2015 which released 0.2 - 0.4 Tg into the UTLS (Bègue et
al., 2017; Reckziegel et al., 2016; Mills et al., 2016). The amounts of $SO_2$ injected during these
events are smaller than those injected during the moderate eruptions which occurred in the
Northern Hemisphere. For instance, the estimation of $SO_2$ reported for these three moderate
eruptions are 10 - 20 times smaller than the Nabro (13.4°N, 41.7°E; Eritrea) in June 2011
(Bourassa et al., 2012; Sawamura et al; 2012), and a quarter of the amount emitted by the



Sarychev (48.1°N, 153.2°E; the Kuril Islands) in June 2009 (Clarisse et al., 2012; Kravitz et al.,
2011; Jégou et al., 2013). This present study focuses on the analysis of the Calbuco eruption.
After 43 years of inactivity, the Calbuco volcano in Chile erupted on the 22 April 2015 followed
by two intense explosive events recorded during the same week. The volcanic plume spread
extensively in the Southern Hemisphere, explained by the dynamical context (Bègue et al.,
2017). Through the analysis of advected Potential Vorticity fields derived for 400 K isentropic
level from MIMOSA (Modèle Isentropique de transport Mésoéchelle de transport de l'Ozone
Stratospherique par Advection) and the Dynamical BArrier Location model (DyBAL; Portafaix
et al., 2003), Bègue et al. (2017) showed that volcanic aerosols are predominantly transported
eastward in planetary-scale tongues. The transport of the volcanic aerosol plume was modulated
by the location of the subtropical barrier and polar vortex, within which most of the zonal
transport took place during the first week following the eruption (Bègue et al., 2017). During
the same year, the Antarctic ozone hole reached a historical record daily average size in
October. The influence of the Calbuco eruption on this significant Antarctic ozone depletion
has been debated. Through the use of Specified Dynamics–Whole Atmosphere Community
Climate Model (SD-WACCM) and balloon observations at Syowa (69.1°S, 34.6° E), Solomon
et al. (2016) found that the Calbuco eruption might be responsible for the extreme ozone
depletion recorded over Antarctic during October 2015. Using the WACCM model in its free-
running configuration, previous works reveal a significant Antarctic ozone column losses
following the moderate Calbuco eruption (Solomon et al., 2016; Ivy et al., 2017). Based on the
use of WACCM model and balloon observations at Syowa, South Pole and Neumayer (70.4°S,
8.2°W), Stone et al. (2017) confirmed the assumption that enhanced ozone depletion was
mainly due to the Calbuco aerosols. Particularly, these stratospheric volcanic aerosols greatly
enhanced austral ozone depletion at 100 - 150 hPa between 55°S and 68°S (Stone et al., 2017).
More recently, Zhu et al. (2018) showed that the Calbuco aerosols depleted around 25% of
ozone near 70°S and created an additional 2.4 million $km^2$ of ozone hole area in September
2015 by using the WACCM model. Conversely, Zuev et al. (2018) supports the assumption that
the stratospheric volcanic aerosols from the moderate magnitude eruption of Calbuco could not
contribute to the intensification of ozone depletion. By combining the ERA-Interim reanalysis
data and the Hybrid Single Particle Lagrangian Intergrated Trajectory (HYSPLIT) model, it
was found that the volcanic plume was outside the stable polar vortex. Zuev et al. (2018)
concluded that the cause of the abnormal stratospheric ozone depletion above Antarctic during
October and November was due to the behavior of the polar vortex in that period. Through the
analysis of the zonal average backscattering from CALIOP, Zhu et al. (2018) showed that the



Calbuco aerosols progressed toward the South Pole at 16 km during June. Moreover, Bègue et
al. (2017) discussed the meridional spread of the Calbuco aerosols toward the South Pole, which
was modulated by the Quasi-Biennal Oscillation.
The present paper reports on the sunphotometry observations of the Calbuco plume at 6 South
American and 3 African sites. The geographical localization of these sites is helpful to improve
the discussion on the latitudinal distribution of the Calbuco plume. Following the Puyehue-
Cordon Caulle eruption, an effort was made to deploy CIMEL sunphotometer system over
Argentina in order to detect aerosols from volcanic ash and Patagonia dust (Otero et al., 2015).
These new databases integrated into the Aerosol RObotic NETwork (AERONET) global
network since 2012 and 2013, will be used and analyzed in this study. The usefulness of the
sunphotometry measurements on the investigation of the aerosols from major and moderate
eruptions has been reported in many previous works (Hobbs et al., 1982; Gooding et al., 1983;
Deshler et al., 1992; Watson and Oppenheimer, 2000, 2001; Porter et al., 2002; Mather et al.,
2004; Sellitto et al., 2017, 2018). Most of these previous works report on the investigation of
optical properties young volcanic plumes near to the source regions. The aims of this study are
to quantify the influence of the Calbuco plume on the total columnar aerosols and to discuss on
the spatio-temporal evolution of the optical properties of the volcanic plume during its transport.
The paper is organized as follows: Section 2 describes the observations and the statistical
approach used to investigate the transport and the optical characteristics of the plume. A
description of the transport of the Calbuco plume is given in Section 3. The statistical detection
of the volcanic plume and its contribution on total columnar aerosols are provided in Section 4.
The discussion on the spatio-temporal evolution of the optical properties of the volcanic plume
during its transport is presented in Section 5. A summary and the perspectives of this study are
given in Section 6.
## 2. Data and methodology
### 2.1 Aerosols and Sulphur dioxide observations
The ground-based sites selected to analyze the transport and the optical characteristics of the
volcanic plume include 9 AERONET sites in south America and southern Africa: Sao Paulo
(23.3°S, 46.4°W), Gobabeb (23.3°S, 15.0°E), Pretoria (25.4°S, 28.1°E), Durban (29.5°S,
31.0°E), Buenos Aires (34.4°S, 58.2°W), Neuquén (38.5°S, 68;0°W), Bariloche (41.0°S,
71.2°W), Comodoro (45.5°S, 67.2°W), and Rio Gallegos (51.3°S, 69.1°W). The localization of
these sites in the Southern Hemisphere allows for a large-scale view of the transport of the
volcanic aerosol plume. Measurements are obtained at 15-min interval under cloud-free and


day time conditions. The direct solar extinction and diffuse sky radiance measurements are used
to compute AOD and to the retrieve aerosol size distribution using the methodology of Dubovik
and King (2000). The estimated uncertainty in AOD measurements under cloud free condition
range from 0.01 to 0.02 (Dubovik et al., 2000, 2006; Eck et al., 2003, 2005). A detailed
description of the CIMEL sunphotometer of the AERONET network and the associated data
retrieval is given by Holben et al. (1998). The AOD values presented in this work are selected
at    Level    2.0    (Cloud-screened    and    quality    assured)    and    downloaded    at:
http://aeronet.gsfc.nasa.gov/. All available observations performed before the Calbuco eruption
until 2016 are also used for this work. The available daily observations and the associated period
for each sitesare reported in Table 1. The available daily observations range from 242 to 3237
at Durban and Buenos Aires, respectively (Table 1). The difference between these sites is
mainly due to their activity period. It is worthy to note that measurements were made quasi-
continuously at each site during the period selected for this study.
During the eruption LiDAR measurements were also performed at the Bariloche site, which is
located nearest the Calbuco volcano (less than 90 km). The LiDAR installed at the Bariloche
airport uses a ND:YAG laser emission system based on Quantel Brilliant B 20 Hz laser, with
366 mJ at 1064 nm (Ristori et al., 2018). The collection of the backscattered photons is done
with a 20 cm Cassegrain telescope connected via an optical fiber to a spectrometric box. The
system can detect the 3 elastics lines: 355, 532 and 1064 nm, two Raman lines 387 and 607 nm,
and water vapor at 408 nm. In this study, elastics LiDAR signals from 532 nm were processed
to retrieve the extinction profile (Fernald, 1984). The Fernald method was applied as the
inversion algorithm, which uses the backscattering-to-extinction ratio called LiDAR-Ratio
(LR) as an input parameter and produce the aerosols extinction profile as output. AOD is
obtained by the integration of the extinction profile across the ranges of interest.
The Moderate Resolution Imaging Spectroradiometer (MODIS) is an instrument aboard the
Terra Earth observation system (EOS AM) and Aqua (EOS PM) satellites. The orbit of Terra
is timed so that it passes over the equator from north to south in the morning. The orbit of Aqua
is timed so that it passes over the equator from north to south in the afternoon. MODIS provides
radiance measurements in 36 spectral bands between 0.44 and 15 $\mu$m, with different spatial
resolution: 250 m (bands 1 and 2), 500 m (bands 3 - 7) and 1 km (bands 8 - 36) (Bennouna et
al, 2013). MODIS aerosols retrievals are done separately over land and ocean using two
independent algorithms (Bennouna et al, 2013). Numerous works such as Kharol et al. (2011),
El-Metwally et al. (2010) and Baddock et al. (2009) have presented a comprehensive
description and operation of the Terra MODIS. The MODIS data used were downloaded at:



https://giovanni.gsfc.nasa.gov/giovanni/. In this study, MODIS AOD data were collected for
2002 - 2016 period over an area of 0.5° x 0.5° latitude and longitude centered on each site to
analyze local as well as regional aerosols loadings. We use high resolution MODIS retrievals
with only very good quality flags to generate AOD statistics over these regions. Taking into
account the conditions mentioned previously the daily observations ranges from 1204 to 3731
at Rio Gallegos and Durban, respectively (Table 1).
The Cloud-Aerosol Lidar with Orthogonal Polarization (CALIOP) onboard the Cloud-Aerosol
Lidar and Infrared Pathfinder Satellite Observation (CALIPSO) was used to study the transport
of the Calbuco plume. CALIPSO flies in a sun-synchronous polar orbit since 2006 with a cycle
of 16 days (Winker et al., 2009). In addition to CALIOP, CALIPSO is composed of two other
instruments: (i) The Imaging Infrared Radiometer (IIR); (ii) The Wide Field Camera (WFC).
CALIOP is an elastically backscattered LiDAR operating at 532 nm and 1064 nm, equipped
with a depolarization channel at 532 nm. Moreover, CALIOP can be categorised into two level
products; level 1 and level 2. The level 1 products are made up of calibrated and geo-located
profiles of the attenuated backscatter returned signal. Level 2 products, on the other hand, are
derived from level 1 products and are classified in three types: profile, vertical feature mask
and layer products (Lopez et al., 2012). Layer products provide layer-averaged properties of
detected aerosol and cloud. Profile products provide retrieved extinction and backscatter
profiles within these layers. The data products are provided at various spatial resolutions. A
detailed description of CALIPSO is given in Winker et al. (2009, 2010 and 2013). In this work
the analysis of the 532 nm aerosol extinction coefficient data product for the period from 23
April to 3 May 2015 was used for the identification of volcanic plumes in the respective days
of observation.
The Ozone Monitoring Instrument (OMI) data product OMSO2G is also used to analyse the
transport of the $SO_2$ plume from the source region to South Africa. OMI is a nadir viewing
spectrometer aboard the National Aeronautics and Space Administration (NASA) EOS AURA
satellite since July 2004. The AURA satellite occupies a near polar sun-synchronous orbit at an
altitude of 705 km (Krotov et al., 2016). A full technical description of the OMI data product is
given in the OMI Algorithm Theorical Basis Document (Barthia et al., 2002). OMSO2G
products used in this work are selected at Level 2.0 with version 3 and are accessible from:
http://disc.sci.gsfc.nasa.gov/Aura/data-holdings/OMI/. The OMSO2G products are obtained
from reflected solar radiation measured in spectral ranges 310-340 nm. The data used in this
work are reprocessed with the new algorithm based on Principal Component Analysis (PCA)
(Li et al., 2013) reducing by half the retrieval noise compared to the previous version. The





OMSO2G products are available for four vertical distributions in sampling grid of 0.125° x
0.125° latitude and longitude (Krotov et al., 2016). The Stratospheric Layer (STL) dataset is
used for this investigation. Earlier works reports that this dataset is suitable for studying
volcanic eruptions (Sangeetha et al., 2018; Li et al., 2013; Krotov et al., 2016). The estimated
uncertainty in $SO_2$ values under cloud free condition for the four vertical distributions is ranging
0.1-0.4 DU and 0.7-0.9 DU at equatorial and high latitudes respectively.
**2.2 Back-trajectory model: HYSPLIT**
The HYSPLIT model was used to calculate backward and forward trajectories in order to derive
information on the transport of the volcanic aerosol plume. The National Oceanic and
Atmospheric Administration (NOAA) Air Resource Laboratory (ARL) developed the
HYSPLIT model that is being used for computing simple and complex air parcel trajectories
(Draxler and Rolph, 2003; Stein et al., 2015). The model can further give information on the
dispersion, chemical transformation and deposition simulations of pollutants. The ability of
HYSPLIT to derive information on the atmospheric transport, dispersion and deposition of
pollutants and volcanic ash has been highlighted in several studies (Stunder et al., 2007; Chen
et al., 2012; Kumar et al., 2017; Sangeetha et al., 2018; Lopes et al., 2019; Shikwambana and
Sivakumar, 2019). A detailed description and its historical evolution is given by Stein et al.
(2015) and is briefly presented here. The calculation of the trajectories is based on hybrid
method between the Lagrangian and Eulerian approaches (Stein et al., 2015). In order to reduce
the uncertainties induced by the meteorological fields and the numerical methods employed,
HYSPLIT can be run in the trajectory clustering mode. The concept of clustering is a
multivariate statistical method which consist in merging the trajectories that are closer to each
other and class them into distinct group. In the present work, the back-trajectories calculations
were helpful to determine if the measured AOD over the selected sites are associated to air
masses that come from the Calbuco volcano. The back-trajectories of air-masses were
calculated every 6h over the selected sites using the Global Data Assimilation System (GDAS)
database for altitudes ranging from 16 to 19 km. The back-trajectories calculations were
performed using the vertical motion calculation method.
**2.3 Methodology**
The characteristics of the optical properties of the volcanic plume were analyzed using AOD
measurements. The AOD measurements are comprised of the volcanic plume $AOD_P$ and the
background values. Thus, the aerosols optical depth of the volcanic plume at a given wavelength





$AOD_P$ ($\lambda$) can be obtained by subtracting the background aerosols optical depth $AOD_B$ ($\lambda$) to
the AOD ($\lambda$) measurements. This methodology has been applied on the Microtops II portable
sunphotometer measurements taken close to volcano site (Watson and Oppenheimer, 2001;
Porter et al., 2002; Mather et al., 2004; Martin et al., 2009; Sellitto et al., 2017, 2018). In this
paper, this approach was applied and adapted to CIMEL sunphotometer measurements taken at
9 sites. To investigate the properties of the volcanic plume, it is important to make background
measurements when the atmosphere is clear of volcanic aerosols as well as measurements
during an eruption. Given that CIMEL sunphotometers are fixed instruments, this requirement
is implicitly respected and allows to define a background situation statistically significant. In
the present work, the background period is defined as the period before January 2015 and after
January 2016. Except for the Durban site, $AOD_B$ ($\lambda$) calculation is based on an average of 6
years of daily observations (Table 1). The anomalies were filtered out in order to obtain daily
optical depth of the "clear" atmospheric layer. Thus, the calculated daily of $AOD_B$ ($\lambda$) means
from April to December were assumed to be within the standard deviation of hourly recorded
data. Moreover, the perturbation induced by the Puyehue Cordon Caulle eruption (40.3°S;
72.1°W, Chile) (Diaz et al., 2014) on the calculation of the background values was taken into
account. As a consequence, the measurements performed during the period ranges from June
to October 2011 were discarded to the calculation of the $AOD_B$ ($\lambda$). Taking into account the
conditions mentioned previously the number of observations used for the calculation of the
daily $AOD_B$ ($\lambda$) means ranges from 89 to 2489 (Table 1). The uncertainty of the in-plume AOD
($\sigma_{AODp}$) is derived by using the standard deviation of the AOD ($\lambda$) measurement and the standard
deviation of the $AOD_B$ ($\sigma_{AOD_B}$), as shown by Sellitto et al. (2017). This uncertainty is given by
equation (1):

$$\sigma_{AOD_P}(\lambda) = \sqrt{\sigma_{AOD}{}^2 + \frac{\left(\sigma_{AOD_B}\right)^2}{n}} \qquad (1)$$

With n the number of individual background measurements $AOD_B$ made to compute the
average background.
The spectral variability of the volcanic plume was analyzed by using the Angström exponent
$\alpha_P$ and the atmospheric turbidity $\beta_P$ (Angström, 1964). The Angström exponent $\alpha_P$ is a well-
known optical proxy for the aerosol size distribution (Shaw, 1983; Tomasi et al., 1997). The
Angström exponent $\alpha_P$ is generally close to zero or negative for aerosols whose extinction
properties are governed by large particles (mean radius distribution greater than 1 μm).





Conversely, $\alpha_P$ values greater than 1 are typical of small particles (mean radius distribution less
than 1 µm). The Angström turbidity $\beta_P$ is the best fit value of AOD$_P$ $(\lambda)$ at 1 µm which depends
on the total number and refractive index of aerosols particles. In this present work, the optical
properties of the volcanic plume were analyzed at three selected wavelength bands which
correspond to ultraviolet (380 nm), visible (500 nm) and near-infrared (1020 nm). Previous
works revealed that the observations in UV wavelengths are helpful for the characterization of
the optical and microphysical properties of the volcanic plume (Porter et al., 2002; Mather et
al., 2004; Sellitto et al., 2017). The Angström exponent $\alpha_P$ and turbidity $\beta_P$ of the volcanic
plume are calculated at selected wavelength pair 380-1020 nm, given by equations (2) and (3):

$$\alpha_P = -\frac{ln\left[\frac{AOD_P(\lambda_1)}{AOD_P(\lambda_2)}\right]}{ln\left[\frac{\lambda_1}{\lambda_2}\right]} \qquad (2)$$

$$\beta_P = AOD_P(\lambda_1).\lambda_1{}^{\alpha_p} \qquad (3)$$

As reported by Sellitto et al. (2017), the use of a spectral interval as large as possible, led to a
decrease in the uncertainties content of $\alpha_P$ and $\beta_P$. According to this research, the uncertainties
of the derived $\alpha_P$ and $\beta_P$ are calculated as follow:

$$\sigma_{\alpha_p} = \left[\frac{1}{ln\left(\frac{\lambda_1}{\lambda_2}\right)}\right]\sqrt{\left(\frac{\sigma_{AOD_P}(\lambda_1)}{AOD_p(\lambda_1)}\right)^2 + \left(\frac{\sigma_{AOD_P}(\lambda_2)}{AOD_p(\lambda_2)}\right)^2} \qquad (4)$$

$$\sigma_{\beta_p} = \lambda_1{}^{\alpha_p}\sqrt{\left(\sigma_{AOD_P}(\lambda_1)\right)^2 + \left(AOD_p(\lambda_1).\ln\lambda_1\right)^2\sigma_{\alpha_p}{}^2} \qquad (5)$$

The spatio-temporal data analysis carried out between MODIS and sunphotometer instruments
helps both to detect the passage of the volcanic plume and its contribution to the total aerosol
column variation, over the selected sites. The methodology described previously, was applied
to the MODIS observations in order to determine the AOD of the volcanic plume AOD$_{P\text{-modis}}$
and the background values AOD$_{B\text{-modis}}$ at 550 nm. MODIS observations are helpful to obtain a
good description of the background behavior over the sites where sampling is not sufficient,
such as the Durban. Table 1, reveals that the MODIS observations used to build the background
situation arehomogenous between the different sites and range between 689 and 3216. It is
necessary to convert the AOD values from MODIS and sunphotometer to a common
wavelength in order to compare them. The AOD values obtained from sunphotometer at 500





nm were converted to the MODIS wavelength following Equation (6) which has already been
used in previous works (Prasad and Singh, 2007; Alam et al., 2011)

$$AOD_{Modis} = AOD_{Photometer} \left(\frac{\lambda_{Modis}}{\lambda_{Photometer}}\right)^{-\alpha} \qquad (6)$$

Where $\alpha$ is the Angström parameter obtained from sunphotometer at 440-870 nm.

## 3. Long-range transport of the volcanic plume

Figure 1a, depicts a time averaged map of OMI STL $SO_2$ column between 22 April and 1 May.
It is clearly shown that $SO_2$ injected into the atmosphere is mainly transported northeastward
over South America and pass over the Sao Paulo site. This figure also reveals a lack of $SO_2$
observations over a region spanning from the vicinity of the Calbuco site to southern parts of
Argentina. This blind region obtained from OMI observations has already been reported in
previous studies as a result of reduced signal-to-noise ratio due to exposure of the low-orbiting
satellite instrumentation to radiation and high energy particles (Fioletov et al., 2016;
Shikwambana and Sivakumar, 2019). Bègue et al. (2017) pointed out a lack of $SO_2$ observations
over this same region by the use of IASI measurements. Over South America, the highest values
of $SO_2$ (2-2.5 DU) are found over Southern Brazil. It can be observed that the outflow of the
$SO_2$ plume towards the Atlantic Ocean is located over the region aforementioned. The plume
is transported by the general circulation over the Atlantic Ocean and reached the African region.
Large values of $SO_2$ (2-3 DU) are mainly observed during the transport of the plume over the
South Atlantic. The $SO_2$ plume entered the western parts of South Africa and spread eastward.
Figure 1a, reveals that $SO_2$ plume was transported over South Africa.
Forward trajectories starting at the Calbuco volcano on 22 April at 14:00 UTC were calculated
for a period of 10-days at three different altitudes between 16 and 20 km (Fig. 1b). It is worthy
to note that the trajectories calculated by HYSPLIT model are in fair agreement with the shape
of the $SO_2$ plume obtained with OMI observations for the same period. Figure 1b, reveals that
the air masses from the Calbuco site leave the South American region on 26 April for the three
selected altitudes. Over the Atlantic Ocean, the spatio-temporal distribution of the air masses
from the Calbuco site depends on the given altitude. The original air masses from the Calbuco
site at 16 km reached the southwestern parts of South Africa from 30 April at around 16.5 km.
These air masses travelled to eastern parts of South Africa one day later and reached the south-
western Indian Ocean on 1 May at 16.9 km. Conversely, the original air masses from the
Calbuco site at 18 and 20 km reached the western and the southern parts of South Africa, on 1



May. Figure 1b, further shows a time averaged map of MODIS AOD at 550 nm between 22
April and 1 May. The area of large AOD values coincide with the forward trajectories. This
could be explained by the presence of volcanic plume. Large values of AOD (0.6 - 1) linked to
the Calbuco eruption are mainly observed over the South American region in the vicinity of the
Neuquén site (Figure 1b). Analysis of the forward trajectories and the MODIS observations,
indicated that the large values of AOD (0.6 - 0.8) around to the African region are in accordance
with the Calbuco plume pathways. Over the Atlantic Ocean, the AOD values associated to the
passage of the Calbuco plume range between 0.4 and 0.8.
The daily extinction coefficients at 532 nm observed by CALIOP between 17:30 UTC to 18:40
UTC on 23 April over the Calbuco volcano and Sao Paulo site are depicted in Figure 2. High
extinction coefficients values (greater or equal to $0.35$ km$^{-1}$) are observed in the vicinity of the
Calbuco volcano and the Neuquén site between 14 and 18 km (Fig. 2a). The back-trajectory
analysis clearly indicates that this thick aerosol layer observed by CALIOP is connected to the
Calbuco eruption. The aerosol plume is structured in two layers separated by weak extinction
coefficients values ($0.01$-$0.02$ km$^{-1}$). The first layer is found between the Calbuco volcano and
the Bariloche site with a vertical extent from 14 to 18 km. The second layer is centered over the
Neuquén site with weaker vertical extent ranging from 15.5 to 18 km. This two-layer structure
of the plume indicates its inhomogeneity at this stage. One day later, the volcanic aerosols layer
is observed between the Neuquén and Buenos Aires sites and structured into one compact layer
extent spanning from 16 to 18 km (Fig. 2b). On 26 April, extinction coefficients values greater
or equal than $0.15$ km$^{-1}$ in link with the Calbuco eruption are observed near to the Sao Paulo
site between 18 and 20 km (Fig. 2c). The spread of the plume over the northeastern parts of
South America is associated with a decrease in thickness. This decrease could be explained by
the sedimentary process which impacted mainly the coarse aerosol particles such as volcanic
ash near to source region. Figure 2, reveals that the top layer aerosols increase slightly during
its transport toward the northeast parts of South America from 23 to 26 April. The feature of
the volcanic aerosol plume over the African region is investigated from the daily extinction
coefficients at 532 nm observed by CALIOP over a region extended from (23° S, 15° E) to
(29°S, 31°E) (Fig. 3). On 30 April, a thin discontinuous aerosols layer (less than 1 km) extended
from west to east is visible on CALIOP observations (Figure 3a). Highest extinction values
($0.05$-$0.07$ km$^{-1}$) are located mainly on the western parts of southern Africa in the vicinity of
the Gobabeb site, whereas weakest values ($0.02$-$0.03$ km$^{-1}$) are observed on the eastern parts in
the vicinity of the Durban site. This suggests that the volcanic plume reached the western parts
of Southern Africa few days later (as shown above) and it followed its propagation by reaching



the eastern side on 30 April. Three days later, an increase of the extinction values and the
thickness of volcanic plume is hence observed in the vicinity of the Durban (Fig. 3b).
Conversely, dilution of the volcanic plume is associated with a decrease by half of the extinction
values over the western parts of South Africa. The observed extinction values over South
America are around 10 times higher than those observed over South Africa (Fig. 2 and Fig. 3).
Overall, the satellite observations indicate that the majority of the aerosol plume is injected in
the lower stratosphere and propagated toward South Africa during the week following the
eruption. It also seems that the southern parts of Argentina were not influenced by the volcanic
plume during this period of time. The synergy between the satellite and ground-based
observations will reinforce the description of the latitudinal distribution of the volcanic plume.
The detection of the volcanic plume from the ground-based observations is discussed in detail
in the next section.
## 4. Influence of the Calbuco plume on the total columnar aerosols
### 4.1 Statistical detection of the volcanic plume
Figure 4 and Figure 5, depict the daily mean evolution of AOD at 550 nm obtained from
sunphotometer and MODIS observations at 6 South American sites between 15 April and 1
December. The selected sites allow for an overview of the latitudinal distribution of AOD over
the South America region. These sites are located in urban and semi-urban areas dominated by
industrial activities and local air pollution (e.g., vehicle emission, air traffic). It appears a north-
south gradient in the background values of AOD obtained from sunphotometer and satellite
observations over South America region (Fig. 4 and Fig. 5). The background values of AOD
and its variability decrease with higher latitudes. The annual mean of AOD background from
Sao Paulo to Rio Gallegos are ranging between $0.11 \pm 4.10^{-3}$ and $0.03 \pm 1.10^{-3}$ respectively.
The highest values of AOD (with annual mean greater or equal than 0.10) and largest variability
are observed at Sao Paulo and Buenos Aires (Figures 4a-b and Figures 4c-d) respectively. These
regions are the most industrialized of the selected Southern American sites (Gassman et al.,
2000; Lopes et al., 2019). The AOD values over the Neuquén site which is located 4 degrees
south to Buenos Aires, is on average half of that those observed, at Buenos Aires (Fig. 5a and
Fig. 5b). In addition to urban and industrial activities, the evolution of the background values
over Sao Paulo and Buenos Aires is influenced by the biomass burning activity which explain
the increase in AOD values during the Austral winter (June-August) (Andreae et al., 2004;
Freitas et al., 2009; Torres et al., 2010). Based on sunphotometer observations over the South
America, Hoelzemann et al. (2009) showed a clear difference on the AOD behavior between



purely fires and urban influenced sites. This explains the contrast between the Northern and
Southern parts of South America. The South American region is influenced by the regional and
long-range transport of air masses from several potential sources of aerosols which induce
seasonal variability on optical properties of aerosols over the country. For instance, the transport
of dust from Patagonia region impacts the seasonal variability on optical properties of aerosols
of the neighbor sites such as Comodoro (Li et al., 2010; Otero et al., 2015). Moreover, the
southern parts of Argentina are frequently impacted by air masses from Antarctic, which could
influence the variability of AOD over Rio Gallegos (Kirchhoff et al., 1997; Otero et al., 2015).
We cannot exclude the hypothesis that the increase of AOD values observed from
sunphotometer and MODIS measurements performed at Rio Gallegos during the Austral winter
could be explained by the transport of air masses from Antarctic region (Figure 5e and Figure
5f). Figure 6a reveals that the background values of AOD and its variability over Sao Paulo and
Buenos Aires compare fairly well with the observed values over the Gobabeb site (in average
$0.10 \pm 6 \cdot 10^{-3}$). The background values of AOD observed at Gobabeb are slightly lower than
those observed at Durban (in average $0.16 \pm 8.10^{-3}$) and Pretoria (in average $0.17 \pm 8.10^{-3}$) both
by sunphotometer and MODIS (Figure 6). The Gobabeb site, in the Namib Desert, is far less
sensitive to the effect of urban pollution than the Durban and Pretoria sites. It can be observed
that all these African sites exhibit an increase of the AOD values during the Austral spring
season which is well-known to be the biomass burning season (Eck et al., 2003; Garstang et al.,
1996; Das et al., 2015; Piketh et al., 1999; Kumar et al., 2017).
The daily AOD measurements performed between 15 April and 1 December 2015 were
compared to background values in order to highlight the passage of volcanic plume.
Furthermore, back-trajectory analysis of daily AOD measurements of 2015 found higher than
background values were made in order to link observations with the Calbuco plume. It is
therefore possible to determine the duration of the plume over a specific site. This duration is
defined as the period during which AOD measurements from 2015 fall outside the standard
deviation of the daily background means. The duration of stay of the Calbuco plume detected
from total columnar aerosols measurements is reported on Table 2 and depicted by the grey
shaded area on Figures 4, 5 and 6. The estimation for the duration of the plume depends mainly
on the availability of daily observations. This condition results in the discrepancies between the
duration of stay between the MODIS and sunphotometer observations. Given its good temporal
resolution, the daily comparison between the observed AOD by MODIS during the background
period and Calbuco event are possible during a long period of time. Measurements collected by
these two instruments are complementary and allow to improve the estimation of the duration





stay over the selected sites. This is clearly illustrated with the case of the Bariloche site where
no measurements were recorded by the sunphotometer from 26 April to 10 May during the
background period (Fig. 3e and Fig. 3f). The use of MODIS observations has allowed to
improve the estimation of the duration stay and conclude that the duration of stay of the plume
was from 23 April to 10 May 2015. In spite of the large variability of the background values
over the Sao Paulo and Buenos Aires sites, the passage of the volcanic plume is clearly visible
from the sunphotometer and MODIS observations (Fig. 4a-b and Fig. 4c-d). This previous
comment is also true for the African sites (Fig. 6). It is worthy to note that the duration of stay
obtained for these sites are in agreement with the chronology reported in the previous subsection
from CALIOP and OMI observations. It can be observed that the passage of the volcanic plume
is not clearly visible over the southern parts of the South American region. Figure 4e reveals
that the passage of the Calbuco plume over the Rio Gallegos site is not visible from AOD
measurements recorded by the sunphotometer. This could be explained by the low daily
sampling during 2015. Conversely, the AOD measurements recorded by MODIS from 12 to 25
May 2015 are higher than daily background values (Fig. 5f). The air masses back-trajectory
calculations confirm the link of these observations with the Calbuco eruption, in agreement
with those reported by Zuev et al. (2018). They analyzed the trajectory of air masses calculated
with the Calbuco volcano from 22 April until the end of August between 15 and 19 km using
the NOAA HYSPLIT model. Zuev et al. (2018) showed that air masses were within limits of
the subtropical stream and polar vortex, impacting the southern parts of the South America.
However, the MODIS observations aforementioned are within the standard deviation of daily
background mean. As a consequence, it is impossible to determine a duration stay over the Rio
Gallegos site following the statistical criteria defined previously. Figures 4e and 4f, do not call
into question the passage of the volcanic plume over the Rio Gallegos site but rather suggest
that the AOD measurements are not statistically significant. A possible explanation for this is
that amount of aerosols transported toward Rio Gallegos are lower than the amounts transported
toward the northern parts of South America which reach South Africa a few days later. The
lower amount of volcanic aerosols gets lost in the variability of the background values. In the
following subsection, the contribution of the volcanic plume on the total columnar aerosols will
be discussed in more details.
**4.2 Statistical variations of the total columnar aerosols**
The daily AOD anomalies induced by the transport of the volcanic plume are estimated and
calculated as a relative difference by considering the daily background as the reference values.



The Bariloche site is clearly the most exposed to the volcanic plume which is illustrated by the
significant difference (in average a factor of 2.5) between the daily AOD measurements of 2015
and background values (Figure 4e). During the first days following the eruption, the AOD
values obtained by LiDAR and sunphotometer observations ranges from 0.18 to 0.24 (Fig. 4e).
This validates that these high values are not as a result of technical artefacts, but attributed to
the passage of the Calbuco plume detected by two ground-based independent instruments. The
maximum values of the relative difference are observed over the Bariloche site (Fig. 7c and
Fig. 7d). Figures 7c and 7d, reveal that the AOD anomalies calculated from the sunphotomoter
and MODIS observations during the first days after the eruption range from 35 to 85 %. Despite
of its geographical distance to the Calbuco volcano site and its significant background
variability, the passage of the plume over the Sao Paulo site has induced a significant daily
AOD anomalies ranging from 20 to 55 % (Fig. 7a and Fig. 7b). The African sites situated in the
west, such as Gobabeb were first impacted by the volcanic plume and AOD anomalies induced
by its spread are significant. Over the Gobabeb site, the daily AOD anomalies range from 10 to
55 % (Fig. 7e and 7f). Table 2, contains the mean values of the AOD anomalies calculated
during the duration stay over all the selected sites. Overall, the AOD anomalies induced by the
passage of the plume over the South American sites are on average higher than those obtained
over Southern Africa sites (Table 2). This is consistent with the contrast on the extinction
coefficient obtained by CALIOP between South America and South Africa reported in the
previous subsection. Table 2, reveals that the highest AOD anomalies (greater than 35 %) are
observed over the northeastern parts of South America, enclosing the Bariloche, Neuquén,
Buenos Aires and Sao Paulo sites. The AOD anomalies for the Comodoro site, which is south
to the Calbuco site, are estimated to at $26.4 \pm 1.5$ % and $14.5 \pm 2.5$ % from sunphotometer and
MODIS, respectively. There seems to be a difference between the sites located to the north and
south of the Calbuco volcano with regards to the AOD anomalies induced by the passage of the
plume. It is worthy to note that latitudinal distribution of AOD anomalies over South America
is consistent with the geographical spread of the volcanic plume obtained by satellite
observations. Both the satellite and ground-based observations reveal that the majority of the
volcanic plume was transported over the northeastern part of South America during the first
days following the eruption. Conversely, the AOD anomalies induced by the spread of the
plume from west to east over South Africa have a homogeneous distribution. On average, the
AOD anomalies for Gobabeb and Durban are estimated at $22.5 \pm 13.0$ % and $24.8 \pm 11.1$ %,
respectively from sunphotometer observations, and estimated at $20.1 \pm 11.2$ % and $27.5 \pm 10.8$
%, respectively from MODIS observations (Table 2). On average, the difference between AOD





anomalies obtained from MODIS and sunphotometer observations is less than 7% with the
exception of the Comodoro site for which the difference is estimated to be 11.9 %. The
discrepancies in term of AOD anomalies between MODIS and sunphotometer may be attributed
primarily to the estimation of the duration stay as previously mentioned. It is important to note
that the discrepancies from background measurements from MODIS and sunphotometer should
not be excluded.
The correlation coefficient and mean bias error (MBE) between sunphotometer and MODIS
AOD observations are depicted in Figure 8 and reported in Table 3. The correlation coefficient
values range from 0.51 to 0.76 with the highest correlation observed over the Pretoria site (Fig.
8c and 8d). This is in agreement with previous studies which reveal that the correlation is
significant between MODIS and sunphotometer over the land instead of over the ocean and
coastal sites due to its low surface reflectivity characteristic (Chu et al., 2002; Vermote et al.,
1997; Hoelzemann et al., 2009; Bréon et al., 2011). It is worthy to note that the correlation
between the sunphotometer and MODIS observations over the Sao Paulo site is similar to those
observed over Durban and Pretoria (Fig. 8a and 8b). In addition, the root mean square error
(RMSE) was calculated and is reported in Table 3. The RMSE and MBE values range from 4.2
% to 13.2 % and from - 9.7 % ± 3.2 % to 8.2 % ± 0.9 %, respectively. The highest discrepancies
between the sunphotometer and MODIS measurements (correlation coefficient lower than 0.60
and RMSE greater than 9%) are observed in the southern parts of South America, close to the
Comodoro and Rio Gallegos sites. In particular, the Comodoro site has the weakest correlation
(0.51) and MBE values of - 9.7 % ± 3.2 % (Fig. 8e and 8f). Previous studies have already
pointed out the bias in AOD data sets collected by MODIS and ground-based instruments in
the Southern Hemisphere between 45°S and 65°S (Zhang and Reid, 2006; Shi et al., 2011;
Lehahn et al., 2010; Madry et al., 2011; Toth et al., 2013). Madry et al (2011) suggested that
this bias could be due to the production of sea-salt particles by the near-surface high winds
occurring along this zonal band. Toth et al. (2013) investigated the quality of MODIS data sets
in this zonal band by comparing them with CALIOP and sunphotometer observations. They
showed about 30-40 % of the observed bias with the ground-based observations could be
attributed to cloud contamination. Table 3, reveals that MODIS overestimates the AOD when
compared to sunphotometer for most part of the selected sites which is consistent with previous
studies (Ichoku et al., 2005; Abdou et al., 2005; Hauser et al., 2005; Hoelzemann et al., 2009).
This bias may be explained by the fact that sunphotometer measurements are made under cloud-
free conditions, whereas MODIS is able to detect aerosols under cloudy pattern. However,
subpixel cloud can be targeted as aerosols which erroneously raise the retrieved AOD





(Hoelzemann et al., 2009). Conversely, we note that MODIS underestimates the AOD when
compare to sunphotometer over the Sao Paulo and the South African sites (Table 3). For the
latter, this was found to be consistent with results obtained by Hao et al. (2005) during the
Southern African Regional Intensive (SAFARI 2000) campaign showing that in the regions of
intense biomass burning, AOD values from MODIS are systematically lower at 470 nm, 550
nm, and 660 nm compared to ground-based measurements by automated and handheld sun
photometers. They suggested that this bias may be due to errors in the assumed aerosol
scattering phase function or surface directional properties. Thus, several potential causes
(surface reflectance, cloud contamination, retrieval bias) could contribute to the discrepancies
between MODIS and sunphotometer observations (Tripathi et al., 2005; More et al., 2013;
Kumar et al., 2015). The statistical results previously mentioned, indicate that the daily AOD
values obtained from sunphotometer and MODIS observations are in fairly good agreement.
Overall, the anomalies induced by the transport of the Calbuco plume can be statistically
detected and evaluated using the AOD measurements at mid-visible wavelengths. In order to
improve the discussion, the observed AOD from the UV to near infrared spectral ranges will be
analyzed in the following section.
**5. Discussion on the optical characteristics of the volcanic plume**
The time evolution of the extinction profile obtained from the collocated LiDAR measurements
at 532 nm confirms the presence of the volcanic plume between 12 and 15 km between 24 and
25 April over the Bariloche site (Fig. 9a). The spectral variability of plume isolated AOD
($AOD_P$) values obtained from sunphotometer observations during the aforementioned period is
shown in Figures 9b and 9c. For both days, the $AOD_P$ evolutions are not characterized by a
wavelength dependence (Fig. 9b) but are similar to those from the UV and NIR spectral range
(average 0.18 ± 0.05). This optical behavior is typical to an aerosol layer dominated by larger
particles such as mineral dust or ash particles (Mather et al., 2004; Bègue et al., 2012; Sellitto
et al., 2018). The Angström exponent $\alpha_P$ and the uncertainty calculated for the wavelength pair
380 - 1020 nm by using Equations (2) and (4) respectively, are also shown in Figure 9b. The
mean values of the Angström exponent $\alpha_P$ on 24 and 25 April are -0.05 ± 0.02 and 0.1 ± 0.06,
respectively. These values confirm a dominance of larger particles with radius greater than 1
μm (Watson and Oppenheimer, 2000). The Angström exponent $\alpha_P$ evolution has a negative
correlation with the $AOD_P$ values. As reported by Sellitto et al. (2018), a negative correlation
indicates a transitory disturbance of relevant burdens of larger particles. Time evolution of the
Angström turbidity $\beta_P$ and its uncertainty derived by using Equations (3) and (5) are depicted



in Figure 9c. The Angström turbidity $\beta_P$ evolution is therefore correlated with the $AOD_P$ values
as depicted in Figure 9b. On average, the Angström turbidity $\beta_P$ values range from $0.16 \pm 0.06$
to $0.19 \pm 0.04$ over the Bariloche site on 24 and 25 April, respectively. For both days, a negative
correlation is observed between the Angström exponent $\alpha_P$ and the turbidity $\beta_P$. This suggests
a significant increase in larger particles over the Bariloche site. These results are found to be in
agreement with estimations of Angström coefficients for ash-bearing plume reported in
previous studies (e.g., Watson and Oppenheimer, 2000, 2001; Porter et al., 2002; Mather et al.,
2004; Martin et al., 2009; Sellitto et al., 2017, 2018). During the minor eruption of the Lascar
volcano (23.4°S, 67.7°W, Chile) in 2003, the Angström exponent $\alpha_P$ (440-1020 nm) and
turbidity $\beta_P$ derived from Microtops sunphotometer observations for bearing-ash were found to
be smaller than 0.3 and ranging from 0.04 to 0.10 respectively (Mather et al., 2004). Through
the use of sunphotometer observations during the last Mount Etna (37.5°N, 14.6°E) eruption,
Sellitto et al. (2017) found the Angström exponent $\alpha_P$ (380-1020 nm) and turbidity $\beta_P$ for
bearing-ash equal to -0.30±0.22 and 0.08±0.05 respectively. During the eruption of Mount
Etna volcano in October 1997, the Angström exponent $\alpha_P$ (440-1020 nm) and turbidity $\beta_P$
derived from CIMEL sunphotometer observations for bearing-ash range from -0.20 to 0.20 and
from 0.16 to 0.65 respectively. It is worthy to note that our Angström coefficients estimations
are in agreement to the ash-bearing plume observed at Mount Etna by Watson and Oppenheimer

19   (2001).

Figure 10 illustrates that the optical characteristics of the volcanic plume evolved during its
transport. On 29 April, the $AOD_p$ values at the Neuquén site are characterized by a wavelength
dependence (Fig. 10a). Figure 10a, indicates that higher $AOD_p$ values are observed in the UV
range (ranging from 0.08 to 0.27) than in the NIR range (ranging from 0.08 to 0.03). This optical
behavior is typical of an aerosol layer dominated by smaller particles (radius lower than 1 µm).
This is confirmed by the Angström exponent $\alpha_P$ values which range from 1.2 to 1.5 (Figure
10a). Figure 10c, reveals that the Angström turbidity $\beta_P$ values range from 0.02 to 0.062 with a
mean value of $0.04 \pm 0.02$ which suggest the presence of a thin aerosol layer. This Angström
coefficients estimation is consistent with ash-free plumes previously observed for other
volcanic eruptions (Watson and Oppenheimer, 2000, 2001; Porter et al., 2002; Mather et al.,
2004; Martin et al., 2009; Sellitto et al., 2017, 2018). For instance, the Angström exponent $\alpha_P$
(440-1020 nm) and turbidity $\beta_P$ estimated from Microtops observations during the Pacaya
volcano (14.2°N, 90.4°W, Guatemala) eruption in 2011 are in average $1.4 \pm 0.7$ and $0.05 \pm$
0.07, respectively. Figure 10b, depicts the $AOD_P$ evolution during the day where the plume
reached the Gobabeb site. Over this site, time evolution of $AOD_P$ is both characterized by an





increase in the volcanic aerosol burden at all wavelengths and a wavelength dependent on 1
May. Thus, higher $AOD_p$ values are observed in the UV range (ranging from 0.04 to 0.15) than
in the NIR range (ranging from 0.03 to 0.05). The Angström exponent $\alpha_P$ range between 0.3
and 1.5 with a mean value of $1.1 \pm 0.75$. This suggests that the particle size distributions of the
plume are not homogenous (Fig. 10b). The increase of the Angström exponent $\alpha_P$ is correlated
with the Angström turbidity $\beta_P$ (ranging from 0.025 to 0.050) (Fig. 10d). These Angström
coefficients values are also consistent with the ash-free plume over the Gobabeb site, on 1 May
(Fig. 10d).
Overall, the ash-bearing plume is characterized by a thick plume containing large particles ($\alpha_P$
$< 0.30$ and $\beta_P > 0.16$) are mainly located near the Calbuco site (such as Bariloche) during the
first days of the eruption (24-25 April). Due to the sedimentation process, these large ash
particles fall out quickly nearby the source region. The fraction that survives to the near-source
fall-out processes are transported over long-range distance. The remote sites to the Calbuco
volcano are hence influenced by this ash-free plume characterized by thin plume composed of
small particle ($\alpha_P \geq 0.3$ and $\beta_P < 0.15$). These results are consistent with previous studies
indicating that the volcanic plume is dominated by larger particles near the source (Hobbs et
al., 1982; Rose et al., 2000; Watson and Oppenheimer, 2000; Webster et al., 2012). Rose et al.
(1982) showed that this phenomenon could be explained by competing mechanisms involving
the adsorption of smaller particles by ash. As reported by Sparks et al. (1997), the aggregation
processes are more important near the source. The evolution of the plume thickness obtained
from sunphotometer observations are consistent with the evolution obtained from CALIOP
observations and presented above in the first subsection. Furthermore, the time evolution of the
optical characteristics of the plume over the selected sites was also analyzed through the
estimation of the Angström coefficients. Their mean values are reported on Tables 4 and 5.
Over the selected sites, the Angström turbidity $\beta_P$ does not evolve significantly in time and on
average is less than 0.06. The Angström exponent $\alpha_P$ tends to decrease slightly in time with
exception of the Sao Paulo site for which the Angström exponent $\alpha_P$ stays fairly constant
(ranging from 1.1 to 1.3). These values are in agreement with the Ansgtröm exponent $\alpha_P$
retrieved from LiDAR observations over Sao Paulo (Lopes et al., 2019). The slight decrease is
clearly visible at the Buenos Aires, Gobabeb and Durban sites for which the Angström exponent
$\alpha_P$ is roughly half and reaches on average a value lower than 0.6 (Tables 4 and 5). These low
Angström coefficient values suggest that the plume evolves with time so that large-particles
dominate the distribution with smaller optical depth. The decrease of the Angström exponent
$\alpha_P$ could be due to enhanced particles growth in the plume induced by microphysical processes





such as aggregation or coagulation. The balance between the growth and removal process
impacts the residence time of the volcanic plume and its size distributions. Moreover, the time
evolution of optical properties of the volcanic plume over the selected sites could be also
explained by the dynamical context. Indeed, the ageing of aerosols plume is a complex
mechanism controlled by many parameters (Bègue et al., 2012; Guermazi et al., 2019).
## 6. Summary and conclusion
The influence of the Calbuco eruption on the total columnar aerosol optical properties over the
Southern Hemisphere has been presented in this investigation. The study focuses mainly on the
analysis of the sunphotometer measurements performed at 6 South American and 3 African
sites. The satellite observations (MODIS, OMI, and CALIOP) were combined with ground-
based observations (sunphotometer and LiDAR). Moreover, the back-trajectory model
(HYSPLIT) was used in order to investigate the transport of the volcanic plume. The spatio-
temporal evolution of the volcanic plume obtained from satellite observations was found to be
consistent with ground-based time series and in agreement with previous works. It is found that
the majority of the plume aerosols were injected up to the lower stratosphere and propagated
towards South Africa during the week following the eruption. The spread of the plume over the
northeastern parts of South America is associated with a decrease in thickness. The satellite
observations pointed out that the southern parts of Argentina are not influenced by the volcanic
plume during the first weeks following the eruption. The synergy between the space-based and
ground-based observations has allowed for further description of the plume over the southern
parts of South America.
The statistical analysis applied on the sunphotometer and MODIS observations pointed out the
presence of a north-south decreasing gradient in the background values of AOD over the South
America region. The highest values of AOD were observed at the Sao Paulo and Buenos Aires
sites which are the most industrialized regions of the selected Southern American sites. The
statistical detection of the plume agreed with the chronology of the plume transport obtained
from satellite observations. Moreover, this statistical approach revealed that the plume has also
impacted the southern parts of South America, albeit in a lesser extent. The anomalies induced
by the transport of the Calbuco plume on the total columnar aerosol optical properties was
statistically evaluated in mid-visible wavelength. The highest AOD anomalies are observed
over the northeastern parts of the South America. Given its proximity to the Calbuco volcano,
the Bariloche site was most impacted by the volcanic plume with daily AOD anomalies ranging
from 35 to 85 %. Over the South African sites, the AOD anomalies induced by the dispersion



of the plume were homogeneously distributed. The observed contrast between the South
America and Africa regions was highlighted by the extinction values from CALIOP
observations.
From the spectral variability of the plume, only AOD was analyzed from the UV to near infrared
spectral ranges. The optical characteristics of the plume near-source region are consistent with
a bearing-ash plume. This spectral analysis of the plume reveals an evolution of its optical
properties over the remote sites to the Calbuco volcano. Thus, the Angström coefficients values
are consistent with an ash-free plume over these sites. The optical evolution of the volcanic
plume during its transport is in agreement with previous works (Hobbs et al., 1982; Rose et al.,
2000; Watson and Oppenheimer, 2000; Webster et al., 2012; Sellitto et al., 2018) and can be
explained by microphysical processes, but also by the dynamical (Baker et al., 2014; Bègue et
al., 2017.; Guermazi et al., 2019; Nimgomba et al., 2019). The Angström coefficients were
useful to obtain a first estimation of the optical characteristics and the size distribution of an
aerosol plume. Nevertheless, the Angström coefficients were not sufficient to describe in detail
the ageing of the aerosols plume during its transport. The parameters that contributed to the
ageing of the Calbuco plume require further investigations which will form the basis for a
forthcoming study.



## Acknowledgements

This work receives funding by the French Centre National de la Recherche Scientifique (CNRS) and the South African National Research Foundation (NRF) through the "Laboratoire International Associé Atmospheric Research in southern Africa and the Indian Ocean" (LIA-ARSAIO, contract n. 78682) and PHC-Protea programmes. This work was supported by the French-Argentine ECOS-Sud A16U01 project. We thank E.Quel, L.Otero, P. Artaxo B. Holben, S. Jacobo, S. Piketh, G. Maggs-Kolling, D. Griffith, S. Venkataraman and their staff for establishing and maintaining the Bariloche, Neuquén, Buenos Aires, Sao Paulo, Comodoro, Rio Gallegos, Gobabeb, Pretoria and Durban sites used in this investigation (http://aeronet.gsfc.nasa.gov/). The authors would like to thank the members of the Servicio Meteorologico Nacional (SMN) for their contribution to data acquisition over South America. The MODIS observations used in this paper were produced with the GIOVANNI online data system, developed and maintained by the NASA GES DISC (https://giovanni.gsfc.nasa.gov/giovanni/). We would like to acknowledge the OMI (http://disc.sci.gsfc.nasa.gov/Aura/data-holdings/OMI) and CALIPSO (https//:eosweb.larc.nasa.gov/) programmes for providing access to data via their websites. The authors especially want to thank the staff members of the CEILAP (Centro de Investigaciones en Láseres y Aplicaciones) team working on LiDAR system at the Bariloche site. The authors gratefully acknowledge the NOAA Air Resources Laboratory (ARL) for the provision of the HYSPLIT transport and dispersion model and/or READY website (http://www.ready.noaa.gov) used in this publication.

## Data availability

The aerosols optical properties from ground-based (sunphotometer) and satellite (MODIS, CALIOP) observations are available on-line from the sources as stated in the manuscript. The sulfur dioxide measurements from OMI observations are available on-line from the sources as stated in the manuscript. The LiDAR data recorded at Bariloche are available from the Servicio Meteorologico Nacional on request.

## Author contribution

N.B analysed the sunphotometer measurements and performed the back-trajectory analysis by the use of HYSPLIT model. L.S contributed to MODIS and CALIOP data analysis and





interpretation. V.S analysed the sulfur dioxide measurements from OMI observations. J.P
contribute to retrieval of aerosol optical properties from LiDAR observations. All authors
contributed to data analysis and interpretation. All authors contributed towards the preparation
of the paper.
**Competing interests**
The authors declare that they have no conflicts of interest.



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





## 2 FIGURES AND TABLES

| Site | Instruments | Period | Daily observation | Background |
|---|---|---|---|---|
| **Gobabeb** | Sunphotometer | 2014-2016 | 694 | 407 |
| (23°S-15°E) | MODIS | 2002-2016 | 2469 | 1954 |
| **Sao Paulo** | Sunphotometer | 2000-2016 | 1957 | 1676 |
| (23°-46°W) | MODIS | 2002-2016 | 2884 | 2369 |
| **Pretoria** | Sunphotometer | 2011-2015 | 1222 | 905 |
| (25°S-28°E) | MODIS | 2002-2016 | 3432 | 2281 |
| **Durban** | Sunphotometer | 2015-2016 | 232 | 89 |
| (29°S-31°E) | MODIS | 2002-2016 | 3731 | 3216 |
| **Buenos Aires** | Sunphotometer | 1999-2016 | 3237 | 2374 |
| (34°S-58°W) | MODIS | 2002-2016 | 3163 | 2648 |
| **Neuquen** | Sunphotometer | 2013-2016 | 678 | 407 |
| (38°S-68°W) | MODIS | 2002-2016 | 3495 | 2980 |
| **Bariloche** | Sunphotometer | 2012-2016 | 463 | 229 |
| (41°S-71°W) | MODIS | 2002-2016 | 2321 | 1806 |
| **Comodoro** | Sunphotometer | 2013-2016 | 830 | 546 |
| (45°S-67°W) | MODIS | 2002-2016 | 3495 | 2980 |
| **Rio Gallegos** | Sunphotometer | 2009-2016 | 1245 | 1000 |
| (51°S-69°W) | MODIS | 2002-2016 | 1204 | 689 |

4 **Table 1.** Number of available and background daily observations at each site for both
5 sunphotometer and MODIS.





| Site | Sunphotometer | | MODIS | |
|---|---|---|---|---|
| | Duration of stay | Anomaly (%) | Duration of stay | Anomaly (%) |
| Gobabeb | 01/05 - 04/05 | 22.5 ±13.0 % | 01/05 - 06/05 | 20.1 ± 11.2 % |
| Sao Paulo | 27/04 - 02/05 | 44.5 ± 24.5 % | 26/04 - 01/05 | 43.4 ± 19.4 % |
| Pretoria | 06/05 - 08/05 | 14.6 ± 5.7 % | 02/05 - 07/05 | 20.7 ± 11.2 % |
| Durban | 04/05 – 08/05 | 24.8 ± 11.1 % | 03/05 - 07/05 | 27.5 ± 10.8 % |
| Buenos Aires | 27/05 – 01/05 | 30.6 ± 5.6 % | 24/05 – 03/05 | 35.5 ± 2.8 % |
| Neuquen | 29/04 | 40.3 ± 0 % | 23/04 – 07/05 | 37.2 ± 15.2 % |
| Bariloche | 24/04 – 25/04 | 53.6 ± 45.1 % | 23/04 – 10/05 | 60.2 ± 30.6 % |
| Comodoro | 29/04 – 30/04 | 26.4 ± 1.5 % | 26/04 – 2/05 | 14.5 ± 2.5 % |

**Table 2.** Averaged anomaly of AOD and corresponding standard deviation (in percentage)
induced by the Calbuco plume during its duration of stay over each site.

| Site | Latitude [°] | $R^2$ | MBE (%) $\frac{1}{n}\sum_{i=1}^{n}\frac{Phot_i - Mod_i}{Phot_i}$ | RMSE (%) $\sqrt{\frac{1}{n-1}\sum_{i=1}^{n}(Phot_i - Mod_i)^2}$ |
|---|---|---|---|---|
| Gobabeb | -23° | 0.69 | 8.2 ± 0.9 | 8.2 |
| Sao Paulo | -23° | 0.70 | 5.2 ± 1.8 | 4.6 |
| Pretoria | -25° | 0.76 | 4.7 ± 1.7 | 8.7 |
| Durban | -29° | 0.71 | 6.9 ± 2.1 | 6.1 |
| Buenos Aires | -34° | 0.68 | -7.1 ± 1.5 | 4.2 |
| Neuquén | -38° | 0.63 | -8.7 ± 2.9 | 7.4 |
| Bariloche | -41° | 0.64 | -7.6 ± 3.3 | 6.3 |
| Comodoro | -45° | 0.51 | - 9.7 ± 3.2 | 13.2 |
| Rio Gallegos | -51° | 0.59 | -8.1 ± 4.2 | 10.4 |

**Table 3.** Statistical parameter for the comparison between Sunphotometer *(Phot)* and MODIS
*(Mod)* observations for each site.





| | Bariloche | | Neuquén | | Buenos Aires | | Sao Paulo | |
|---|---|---|---|---|---|---|---|---|
| | $\alpha_P$ | $\beta_P$ | $\alpha_P$ | $\beta_P$ | $\alpha_P$ | $\beta_P$ | $\alpha_P$ | $\beta_P$ |
| 24th April | -0.01 (±0.02) | 0.16 (±0.02) | | | | | | |
| 25th April | 0.02 (±0.05) | 0.19 (±0.02) | | | | | | |
| 26th April | | | | | | | | |
| 27th April | | | | | 1.6 (±0.6) | 0,01 (±0.06) | 1.3 (±0.2) | 0.04 (±0.03) |
| 28th April | | | | | 0,8 (±0,3) | 0.01 (±0.005) | 1.1 (±0.4) | 0.04 (±0.02) |
| 29th April | | | 1.4 (±0.1) | 0.04 (±0.02) | | | 1.2 (±0.2) | 0.02 (±0.6 |
| 30th April | | | | | 0.87 (±0.09) | 0.05 (±0.006) | | |

**Table 4.** Mean and standard deviation for the plume-isolated Angström exponent and turbidity
during the 24th-30th April for the Bariloche, Neuquén, Buenos Aires and Sao Paulo. These
values are obtained from sunphotometer measurements. The grey grids indicate that there were
no observations.

| | Gobabeb | | Pretoria | | Durban | |
|---|---|---|---|---|---|---|
| | $\alpha_P$ | $\beta_P$ | $\alpha_P$ | $\beta_P$ | $\alpha_P$ | $\beta_P$ |
| 01st May | 0.35 (±0.9) | 0.03 (±0.007) | | | | |
| 02nd May | 0.42 (±0.06) | 0.16 (±0.006) | | | | |
| 03rd May | 1.1 (±0.75) | 0.06 (±0.005) | | | | |
| 04th May | 0.87 (±0.11) | 0.01 (±0.008) | | | 0.55 (±0.14) | 0.05 (±0.02) |
| 05th May | | | | | 0.72 (±0.36) | 0.02 (±0.006) |
| 06th May | | | 1.5 (±0.41) | 0.004 (±0.002) | 1.1 (±0.30) | 0.04 (±0.005) |
| 07th May | | | 1.2 (±0.40) | 0.02 (±0.01) | 1.85 (±0.85) | 0.014 (±0.012) |
| 08th May | | | 0.5 (±0.30) | 0.09 (±0.01) | 1.18 (±0.23) | 0.02 (±0.01) |

**Table 5.** Mean and standard deviation for the plume-isolated Angström exponent and turbidity
during the 01st-08th May for the Gobabeb, Pretoria and Durban. The grey grids indaicate that
there were no observations.



2 **Figure 1:** Time averaged maps of (a) SO$_2$ column in the lower stratosphere observed by OMI
3 and (b) AOD observed by MODIS during the 22 April-1 May period. Forward-trajectories





analysis of air masses starting at the Calbuco volcano coordinates at 16 km (dots), 18 km
(diamonds) and 20 km (triangles) are plotted in white lines. The location of the selected sites
are indicated by: (B) Bariloche, (N) Neuquén, (BA) Buenos Aires, (SP) Sao Paulo, (C)
Comodoro, (R) Rio Gallegos, (G) Gobabeb, (D) Durban, (P) Pretoria.





**Figure 2:** Daily zonal extinction coefficient (km$^{-1}$) at 532 nm observed by CALIOP over the Calbuco volcano and in the vicinity of the Sao Paulo site (23°S, 46°W) on (a) 23 April, (b) 24 April and (c) 26$^{th}$ April. The red star and the blue square correspond to the localization of the Calbuco volcano and the maximum extinction values respectively. Back-trajectory analysis





1   between the maximum extinction values and the Calbuco volcano are plotted by the green

2   curve.

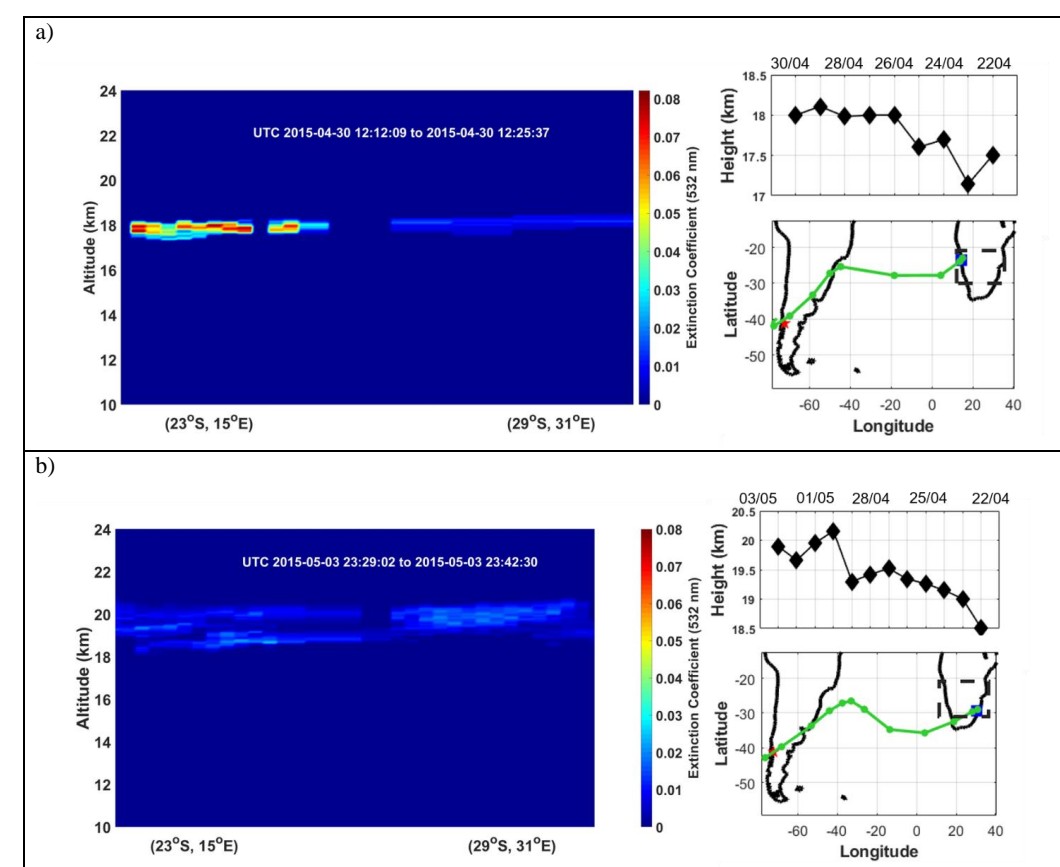

**Figure 3:** Daily zonal extinction coefficient (km⁻¹) at 532 nm observed by CALIOP over the
South African region on (a) 30 April and (b) 3 May. The red star and the blue square correspond
to the localization of the Calbuco volcano and the maximum extinction values respectively.
Back-trajectory analysis between the maximum extinction values and the Calbuco volcano is
represented by the green curve.









**Figure 4:** Daily mean of AOD (550 nm) obtained over (a,b) Sao Paulo, (c,d) Buenos Aires and
(e,f) Bariloche obtained from Sunphotometer and LiDAR (on the left panel) and from MODIS
observations (on the right panel) from 15 April to 1 December. The grey area corresponds to
the influence of the volcanic plume over a given site. The black line indicates the monthly mean
values.







**Figure 5:** Daily mean of AOD (550 nm) obtained over (a,b) Neuquén, (c,d) Comodoro and (e,f)
Rio Gallegos obtained from Sunphotometer (on the left panel) and from MODIS observations
(on the right panel) from 15 April to 1 December. The grey area corresponds to the influence
of the volcanic plume over a given site. The black line indicates the monthly mean values.





**Figure 6:** Daily mean of AOD (550 nm) obtained over (a,b) Gobabeb, (c,d) Pretoria and (e,f)
Durban obtained from Sunphotometer (on the left panel) and from MODIS observations (on



1    the right panel) from 15 April to 1 December. The grey area corresponds to the influence of the

2    volcanic plume over a given site. The black line indicates the monthly mean values.









1 **Figure 7:** Daily mean of AOD anomaly (%) at (a,b) Sao Paulo, (c,d) Bariloche and (e,f)

2 Gobabeb calculated from Sunphotometer (left panel) and MODIS (right panel) observations

3 between 19 April and 31 May.







**Figure 8:** Correlation of AOD daily mean observations (@550 nm) between Sunphotometer
and MODIS during all the period of available data at (a,b) Sao Paulo; (c,d) Pretroria and (e,f)
Comodoro. The histograms shows the mean bias error (MBE) between the two datasets by the
number of observations for each site.







**Figure 9:** (a) Time evolution of the extinction profile (@532 nm) obtained from LiDAR
observations at Bariloche between 24 and 25 April 2015. (b) Time evolution of the plume-
isolated AOD from UV to NIR versus the Angström exponent (380-1020 nm) from
Sunphotometer observations at Bariloche between 24 and 25 April 2015. (c) Time evolution of
the Angström turbidity versus the Angström exponent (380-1020 nm) from Sunphotometer
observations at Bariloche between 24 and 25 April 2015.





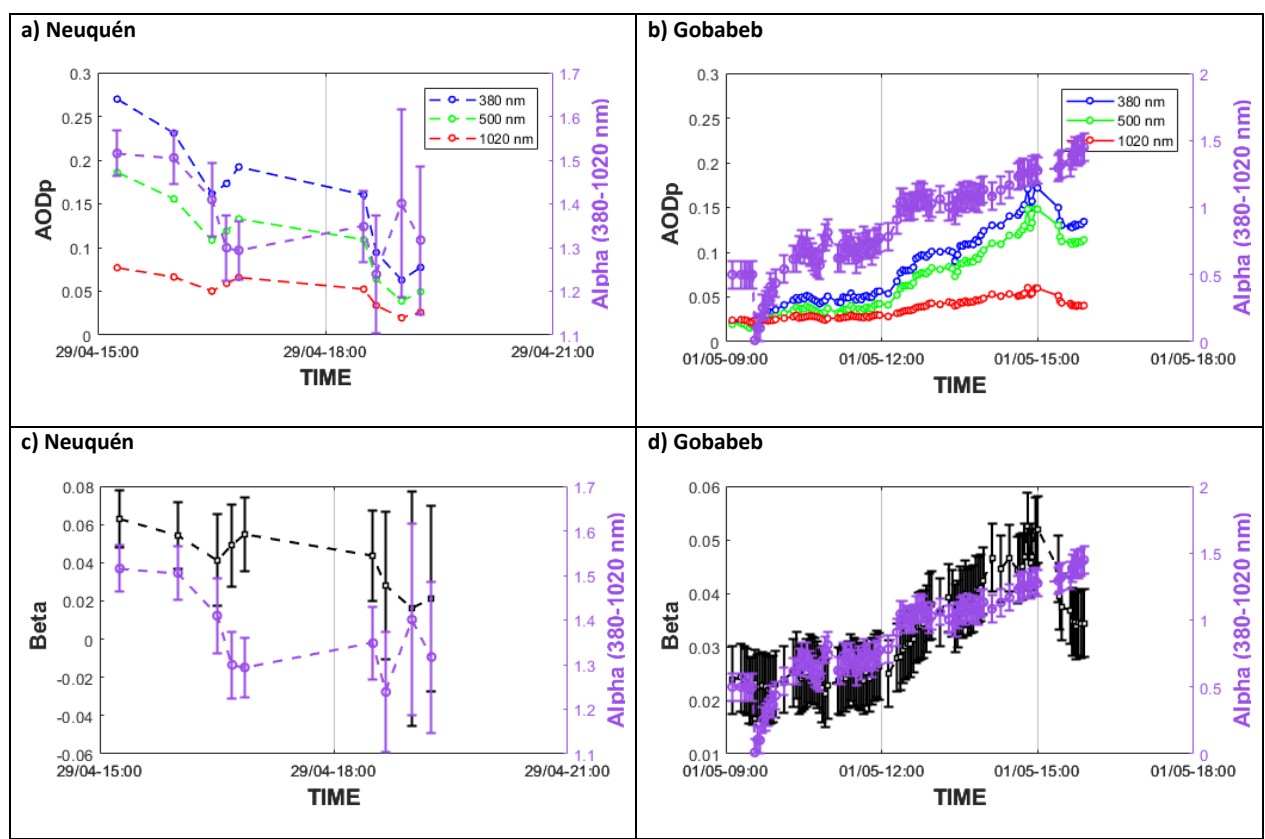

**Figure 10:** Time evolution of the plume-isolated AOD from UV to NIR versus Angström
exponent (380-1020 nm) (a) for 29 April over Neuquén and (b) for 1 May over Gobabeb. Time
evolution of Angström turbidity versus Angström exponent (380-1020 nm) (c) for 29 April over
Neuquen and (d) for 1 May over Gobabeb.

