# Peer review of "volcanic eruption from ground-based and space-borne observations"

_Annales Geophysicae, 2019_

## Referee Comment (RC1) · Anonymous Referee #1 · 19 Dec 2019

The papers covers the statistical analysis on the optical properties of the "plumes" generated after the Calbuco Eruption event and describes how it developed on its way from South America towards the South African Continent. The data provided by CALIPSO, OMI and MODIS satellites and other model analysis were used to enhance the quality of this study. At some point on its trajectories the plumes could be probed by ground based stations namely sunphotmeters and lidars were used to charecterize the geometrical and/or optical properties. As a wrap-up procedures the data can be used as input to regional models such as MIMOSA. The quality of presentation is very clear and well performed. In this sense I would accept the paper in its present form with the only suggestion to use the acronym "LiDAR" as a noun "lidar"similarly to the

word radar. But this is up to the authors to consider.

---

## Referee Comment (RC2) · Anonymous Referee #2 · 30 Dec 2019

Statistical analysis of the long-range transport of the 2015 Calbuco volcanic eruption from ground-based and space-borne observations.

In this study the authors presented results based on statistical analysis of the influence of the 2015 Calbuco eruption (Chile) on the total columnar aerosol optical properties over the Southern Hemisphere. In order to investigate the aerosol optical properties in the middle and upper trophosphere injected by the Calbuco Volcano statistical analysis were applied to AERONET sunphotometer database at six stations of South America and three at African Continent. The analysis consisted on the retrieval of the AOD anomalies calculated by the relative difference of the daily AOD background as the

reference values, for both sunphotometer and MODIS instrument. The transport and the spatio-temporal evolution of the volcanic plume were investigated using satellite data and air-masses back-trajectory model, allowing the increasing on the quality of the analyzes. On an overall, the study is clear, well presented and discussed. The study will contribute significantly to the understanding of the volcanic plumes transportation around the globe and all over South America region. Therefore, I would like to recommend the publishing of the manuscript after some revisions.

Major comments and suggestions:

Page 7 – line 1 to 6 – Are the authors using MODIS on board of Terra or Aqua satellite?

Page 11-line 21 to page 12-line 8 The discussion based on results presented on figure 1 are a bit confusing. Please, could the authors discuss in more details about $SO_2$ results from OMI, and mainly, a more detailed discussion about the retrieval of the air-masses trajectories presented on Fig. 1b? The authors also could enumerate as figure 1a), figure 1b and figure 1c), since the three of them are from a distinct method of retrieval. Please consider increase the quality of the figures since it is very difficult for the readers identify all the sites presented on figure 1a).

Page 12 – line 20 – the authors could consider increase the quality and the size of the figure 2. Please, consider increase the axis font size of the Extinction coefficient from CALIPSO data, and include a more detailed map of the site. Please consider also include the CALIPSO overpass trajectories, it could increase the understating and the visualization of the volcanic plume transportation all over the South America.

Page 12 – lines 20 to 22 – "On 26 April, extinction coefficients values greater or equal than 0.15 km-1 in link with the Calbuco eruption are observed near to the São Paulo site between 18 and 20 km (Fig. 2c)." The reference J. S. Lopes, F.; Silva, J.J.; Antuña Marrero, J.C.; Taha, G.; Landulfo, E. Synergetic Aerosol Layer Observation After the 2015 Calbuco Volcanic Eruption Event. Remote Sens. 2019, 11, 195. Discussed in detail the aerosol optical properties of Calbuco's plume over São Paulo using lidar and

[Figure]

CALIPSO data. Please, consider using this as reference.

Page 16 – lines 3 to 4 - the authors declare, "During the first days following the eruption, the AOD values obtained by LiDAR and sunphotometer observations ranges from 0.18 to 0.24 (Fig. 4e)". It is not clear how the AOD values using the Bariloche lidar data were retrieved. It was using the Raman signal providing independent values of backscatter and extinction profiles of Calbuco ashes plume or applying Klett-Fernald-Sasano Method (KFS), based on AOD from AERONET? If the second case was applied, what is the error considered since the AOD from AERONET are retrieved by the total aerosol column and lidar can provide the AOD from a single aerossol plume? The AOD used in the KFS Method are the plume isolated AODp? The authors considered using other approach to retrieve the AOD from plume using the lidar data?

Section 5 – page 18 and 19 – it is not so clear the relation of Angström turbidity and Angström exponent, neither the Angström turbidity and the AOD variation. Please, consider discuss this point in more detail.

Minor comments and suggestions:

Please, consider increase the quality, the resolution and also the size of all figures. Please consider performing a complete typing revision, figure enumeration and citation. In addition, a complete revision on the citations throughout the text and in the references section.

Page 7 – line 17 – please correct the reference citation Lopez et al., 2012 – to Lopes et al., 2012

Please, consider correct the following reference: F. J. S. Lopes, G. L. Mariano, E. Landulfo and E. V. C. Mariano (September 12th 2012). Impacts of Biomass Burning in the Atmosphere of the Southeastern Region of Brazil Using Remote Sensing Systems, Atmospheric Aerosols - Regional Characteristics - Chemistry and Physics, Hayder Abdul-Razzak, IntechOpen, DOI: 10.5772/50406. Available from:

https://www.intechopen.com/books/atmospheric-aerosols-regional-characteristics-chemistry-and-physics/impacts-of-biomass-burning-in-the-atmosphere-of-the-southeastern-region-of-brazil-using-remote-sensi

Page 10 – line 24 – Please, correct "are homogenous" sentence.

Page 15 – line 3 – the authors should mention figures 5e and 5f instead of fig 3e and 3f.

Page 15 – line 23 – the authors should mention figures 5e and 5f instead of figure 4e and 4f

Please, consider correct the following reference: J. S. Lopes, F.; Silva, J.J.; Antuña Marrero, J.C.; Taha, G.; Landulfo, E. Synergetic Aerosol Layer Observation After the 2015 Calbuco Volcanic Eruption Event. Remote Sens. 2019, 11, 195.

---

## Author Comment (AC1) · 24 Jan 2020

First of all, the authors acknowledge the referee and the editor for the time spent to review this manuscript and also for their constructive comments. The modifications are indicated by italic and red bold fonts in the revised manuscript.

REFEREE 1 RC1 : In this sense I would accept the paper in its present form with the only suggestion to use the acronym "LiDAR" as a noun "lidar"similarly to the word radar. But this is up to the authors to consider.

Authors: The suggestion mentioned by the referee 1 was included in the revised

manuscript.

Please also note the supplement to this comment:
https://www.ann-geophys-discuss.net/angeo-2019-138/angeo-2019-138-AC1-supplement.pdf

---

## Author Comment (AC3) · 24 Jan 2020

**RESPONSES TO REFEREE**

First of all, the authors acknowledge the referee and the editor for the time spent to review this manuscript and also for their constructive comments. The modifications are indicated by italic and red bold fonts in the revised manuscript.

**REFREE 2**

**Major revisions:**

**1) Page 7, line 1 to 6: Are the authors using MODIS on board of Terra or Aqua satellite?**
**Authors:** We used the MODIS aboard the Terra (EOS AM) satellite. This information was added in the revised manuscript.

**2) Page 11, line 21 to page 12, line 8: The discussion based on results presented on figure 1 are a bit confusing. Please, could the authors discuss in more details about SO2 results from OMI, and mainly, a more detailed discussion about the retrieval of the air-masses trajectories presented on Fig. 1b? The authors also could enumerate as figure 1a), figure 1b and figure 1c), since the three of them are from a distinct method of retrieval. Please consider increase the quality of the figures since it is very difficult for the readers identify all the sites presented on figure 1a).**
**Authors:** This part of Section 3 was re-written by improving the discussion on the $SO_2$ results and the air masses trajectories (See revised manuscript). The synergy between the forward trajectories calculated from HYSPLIT model and MODIS observations allows to determine the AOD values in link to the Calbuco eruption. It is for this reason that the authors have previously decided to superimpose the forward trajectories on the time averaged map of MODIS AOD (550 nm). In order to reduce the confusion, Figure 1 was re-plotted following the suggestion of the referee 2: (a) Time averaged map of $SO_2$ column in the lower stratosphere observed by OMI during the 22 April-1 May period; (b) Forward-trajectories analysis of air masses from HYSPLIT model starting at the Calbuco volcano coordinates at 16 km, 18 km and 20 km; (c) Time averaged map of MODIS AOD (550 nm) during the 22 April-1 May period. Furthermore, the localization of the selected sites are now indicated by white boxes and their initials in the purpose to have easier identification of them. It is worthy to note that the quality of the Figure 1 was improved in the revised manuscript. *(See page 6-7 of this document)*

**3) Page 12, line 20: the authors could consider increase the quality and the size of the figure 2. Please, consider increase the axis font size of the Extinction coefficient from CALIPSO data, and include a more detailed map of the site. Please consider also include**

**the CALIPSO overpass trajectories, it could increase the understating and the visualization of the volcanic plume transportation all over the South America.**

Authors: Figure 2 was re-plotted following the suggestions of the referee 2. Thus, the CALIPSO overpass trajectories and the localization of the site were included (See revised manuscript). Furthermore, Figure 3 was also re-plotted following these suggestions. *(See page 8-10 of this document)*

**4) Page 12, lines 20 to 22: "On 26 April, extinction coefficients values greater or equal than 0.15 km-1 in link with the Calbuco eruption are observed near to the São Paulo site between 18 and 20 km (Fig. 2c)."**

**The reference J. S. Lopes, F.; Silva, J.J.; Antuña Marrero, J.C.; Taha, G.; Landulfo, E. Synergetic Aerosol Layer Observation After the 2015 Calbuco Volcanic Eruption Event. Remote Sens. 2019, 11, 195. Discussed in detail the aerosol optical properties of Calbuco's plume over São Paulo using lidar and CALIPSO data. Please, consider using this as reference.**

Authors: As suggested by the referee 2 the discussion on the optical properties of the Calbuco plume over São Paulo was improved in the revised manuscript by considering Lopes et al (2019) as reference.

**5) Page 16, lines 3 to 4: the authors declare, "During the first days following the eruption, the AOD values obtained by LiDAR and sunphotometer observations ranges from 0.18 to 0.24 (Fig. 4e)". It is not clear how the AOD values using the Bariloche lidar data were retrieved. It was using the Raman signal providing independent values of backscatter and extinction profiles of Calbuco ashes plume or applying Klett-Fernald-Sasano Method (KFS), based on AOD from AERONET? If the second case was applied, what is the error considered since the AOD from AERONET are retrieved by the total aerosol column and lidar can provide the AOD from a single aerossol plume? The AOD used in the KFS Method are the plume isolated AODp? The authors considered using other approach to retrieve the AOD from plume using the lidar data?**

Authors: We thank the referee 2 for this relevant comment. In this study, elastics LiDAR signals from 532 nm were processed to retrieve the extinction profile (Fernald, 1984[1]). The Klett-Fernald-Sassano (KFS) method was applied as the inversion algorithm which has been
* * *
[1] Fernald, F.G. Analysis of atmospheric lidar observations: some comments. *Appl. Opt.*, 23, 652–653, 1984

applied to lidar signal and take into account both earlier Klett and Fernald approaches together. Our future objectives also to use a single method of approach during a measurement campaign. The retrieval of aerosol optical properties from an elastic lidar signal is not easy due to the fact that the elastic lidar signal has 2 unknown: aerosols backscatter and extinction coefficients. The analytical solution obtained from the KFS method assumes a constant relation between the extinction-to-backscattering profiles, named lidar-ratio (LR) which is a key point of this method. The LR value is obtained from the values reported in the literature in link with the nature of the aerosols (Trickl et al., 2013[2] ; Ridley et al., 2014[3] ; Sakai et al., 2016[4] ; Khaykin et al., 2017[5]) or iteratively by using a reference AOD given by Sunphotometer. In this study, we used the AOD given by the sunphotometer deployed at the Bariloche site. Another parameter that we need to retrieve the optical properties is the altitude reference which correspond to altitude without aerosols load. The statistical uncertainties of the optical products are calculated based on a Monte Carlo method (D'Amico et al, 2015[6]), widely used in Earlinet Network. The systematics errors related to the inversion method are mainly related to the inputs parameters: LR and altitude reference values. On average, errors related to altitude reference is 15%, and those related to LR around 20%. This discussion was included in the revised manuscript.

When the volcanic plume is detected, its optical depth can be obtained from the raw lidar signal without any assumption (Young 1995, 2005). It is worthy to note that clear-air region above
* * *
[2] Trickl et al (2013) : 35yr of stratospheric aerosol measurements at Garmisch-Partenkirchen: from Fuego to Eyjafjallajökull, and beyond, Atmos. Chem. Phys., 13, 5205–5225, doi:10.5194/acp-13-5205-2013

[3] Ridley et al (2014).:Total volcanic stratospheric aerosol optical depths and implications for global climate change, Geophys. Res. Lett., 41, 7763–7769, doi:10.1002/2014GL061541

[4] Sakai et al (2016).: Long-term variation of stratospheric aerosols observed with lidars over Tsukuba, Japan, from 1982 and Lauder,New Zealand, from 1992 to 2015, J.Geophys.Res.Atmos., 121,10283–10293, doi:10.1002/2016JD025132, 2016.

[5] Khaykin et al (2017). Variability and evolution of the midlatitude stratospheric aerosol budget from 22 years of ground-based lidar and satellite observations. *Atmospheric Chemistry and Physics*, *17*(3), 1829-1845.

[6] D'Amico et al. (2015). "EARLINET Single Calculus Chain – technical – Part 1: Pre-processing of raw lidar data". Atmos. Meas. Tech. Discuss., 8, 10387–10428, 2015. www.atmos-meas-tech-discuss.net/8/10387/2015/. doi:10.5194/amtd-8-10387-2015

and below the aerosol layer is needed to retrieve the AOD of the plume. CALIOP observations reveal that the aerosols plume has a vertical extent from 14 to 18 km the day following the eruption with maximum values of extinction located at 17 km (Figure 2, revised manuscript). The lidar signal recorded at the Bariloche site is ranging from ground to 15 km. As a consequence, the main part of the aerosol layer was not observed by the lidar at the Bariloche site. It is therefore difficult to retrieve properly the AOD from plume using these lidar profiles. It is for this reason that the author did not discuss in details on extraction of $AOD_P$ from these lidar observations.

**6) Section 5, page 18 and 19: it is not so clear the relation of Angström turbidity and Angström exponent, neither the Angström turbidity and the AOD variation. Please, consider discuss this point in more detail.**

**Authors:** As suggested by the referee 2 this part of the discussion was improved by including more interpretation on the relation between the Angström parameters and the AOD variation. The Angström exponent $\alpha_P$ and Angström turbidty $\beta_P$ are linked by the empirical Angström law (Eq. 1) :

$$AOD_P(\lambda) = \beta_P \, \lambda^{-\alpha_P} \ (Eq.\,1)$$

The $\alpha_P$ parameter characterizes the spectral features of aerosol and its mainly related to the size of the particles while the $\beta_P$ parameter is related to the particle concentration and represents the AOD at 1 µm. The $\beta_P$ parameter increases with the number of particles. As a consequence, weak $\beta_P$-values is consistent with the plume being less opaque and weak AODp values. Observations of bigger $\beta_P$ (greater than 0.1: thick aerosol layer) and smaller $\alpha_P$ (less than 0.5: coarse particle) can be associated to relevant burdens of large particles, like for ash puff. Conversely, observations of weaker $\beta_P$ (from 0.001 to 0.1: less opaque plume) and higher $\alpha_P$ (greater than 0.5: fine particle) can be associated to ash-free.

**Minor revisions:**

**7) Please, consider increase the quality, the resolution and also the size of all figures. Please consider performing a complete typing revision, figure enumeration and citation. In addition, a complete revision on the citations throughout the text and in the references section.**

**Authors:** We understand the point of the view of the referee 2 and also the importance of the quality of the Figures for an article. . As suggested we have replotted the figures in a better resolution in the revised manuscript.

**8) Page 7 – line 17 – please correct the reference citation Lopez et al., 2012 – to Lopes et al., 2012**

**Please, consider correct the following reference: F. J. S. Lopes, G. L. Mariano, E. Landulfo and E. V. C. Mariano (September 12th 2012). Impacts of Biomass Burning in the Atmosphere of the Southeastern Region of Brazil Using Remote Sensing Systems, Atmospheric Aerosols - Regional Characteristics - Chemistry and Physics, Hayder Abdul-Razzak, IntechOpen, DOI: 10.5772/50406. Available from: https://www.intechopen.com/books/atmospheric-aerosols-regional-characteristicschemistry-and-physics/impacts-of-biomass-burning-in-the-atmosphere-of-thesoutheastern-region-of-brazil-using-remote-sensi**

Authors : This typo error was corrected and the reference was included in the revised manuscript.

**9) Page 10, line 24: Please, correct "are homogenous" sentence.**

Authors : It was corrected in the revised manuscript.

**10) Page 15, line 3: The authors should mention figures 5e and 5f instead of fig 3e and 3f.**

Authors:  It was corrected in the revised manuscript.

**11) Page 15, line 23: The authors should mention figures 5e and 5f instead of figure 4e and 4f**

Authors: The referee 2 is right. As a consequence, this point was clarified in the revised manuscript.

**12) Please, consider correct the following reference: J. S. Lopes, F.; Silva, J.J.; Antuña Marrero, J.C.; Taha, G.; Landulfo, E. Synergetic Aerosol Layer Observation After the 2015 Calbuco Volcanic Eruption Event. Remote Sens. 2019, 11, 195.**

Authors: The correction was added to the revised manuscript.

[Figure]

c)

[Figure]

*Figure 1: (a) Time averaged map of SO₂ column in the lower stratosphere observed by OMI during the 22 April-1 May period. (b) Forward-trajectories analysis of air masses from HYSPLIT model starting at the Calbuco volcano coordinates at 16 km (blue line with grey dots), 18 km (green line with grey diamonds) and 20 km (red line with grey triangles). (c) Time averaged map of MODIS AOD (550 nm) during the 22 April-1 May period. The localization of the selected sites are indicated by black boxes and their initials: (B) Bariloche, (N) Neuquén, (BA) Buenos Aires, (SP) Sao Paulo, (C) Comodoro, (R) Rio Gallegos, (G) Gobabeb, (D) Durban, (P) Pretoria.*

[Figure]

*Figure 2: Daily zonal extinction coefficient (km⁻¹) at 532 nm observed by CALIOP over the Calbuco volcano and in the vicinity of the Sao Paulo site (23˙S, 46˙W) on (a) 23 April, (b) 24 April and (c) 26ᵗʰ April. The red star and the blue square correspond to the localization of the Calbuco volcano and the maximum extinction values respectively. Back-trajectory*

*analysis between the maximum extinction values and the Calbuco volcano are plotted by the green curve. The CALIPSO overpass trajectories are plotted by the orange curve*

[Figure]

***Figure 3: Daily zonal extinction coefficient (km⁻¹) at 532 nm observed by CALIOP over the South African region on (a) 30 April and (b) 3 May. The red star and the blue square correspond to the localization of the Calbuco volcano and the maximum extinction values respectively. Back-trajectory analysis between the maximum extinction values and the Calbuco volcano is represented by the green curve.***